# Electron Emission Cross Section from Methane under 250 keV Proton Impact

**Debasmita Chakraborty [1], László Gulyás [2] and Lokesh C. Tribedi [1],***

[1] Deaprtment of Nuclear and Atomic Physics, Tata Institute of Fundamental Research, Mumbai 400005, India
[2] Institute for Nuclear Research (ATOMKI), H-4001 Debrecen, Hungary
\* Correspondence: lokesh@tifr.res.in

**Abstract:** We measure double differential cross sections (DDCS) of electrons emitted from $CH_4$ molecules in collisions with 250 keV protons. The projectile ions are obtained from a 400 kV electron cyclotron resonance-based ion accelerator (ECRIA). We study the energy and angular distributions of the electron DDCS. The observed double and single differential and the total cross section are compared with the state-of-the-art continuum distorted wave eikonal initial state (CDW-EIS) model predictions. Two different approaches are used considering the different target descriptions: complete neglect of differential overlap (CNDO) and molecular orbital (MO) approximations. The MO model uses two different scaling parameters ($d = 0.7$ and $1.0$). In the energy distribution of the DDCS, the carbon KLL Auger line is also observed at 240 eV. The single differential cross section (SDCS) and total cross section (TCS) are derived. Both the MO-based CDW-EIS models are in good agreement with the experimental results; however, the CNDO approach overestimates the data.

**Keywords:** e-emission; e-spectrometer; DDCS; ECR source; protons; CDW-EIS





## 1. Introduction

The ionization of atoms and molecules under charged particle impact is an important aspect in the study of various features of ion–atom or ion–molecule collisions in general. Methane, ($CH_4$) a simple tetrahedral hydrocarbon molecule with four equivalent C-H bonds is present in earth as well as in the interstellar medium (ISM) [1,2]. The collisions with methane also play an important role as a reference for the collisional aspects involving PAHs and DNA-bases [3,4]. Some studies have been reported on electron emission from methane using electrons, protons and different charged particles [5–12].

Apart from the ionization study, electron capture from methane molecules has also been investigated by several groups [13–16]. Studies are present in the literature on the fragmentation of methane [17–24]. However, systematic studies of the ionization of methane using protons and detailed comparisons of the experimental data with different theoretical approaches are very rare.

In this work, scattered electron emissions from $CH_4$ were studied for a range of electron energies and different scattering angles in collisions with 250 keV protons. A comparison with the CDW-EIS model calculations using the complete neglect of differential overlap (CNDO) and molecular orbital (MO)-based target wave functions [25–30] is presented, which was missing in most of the earlier studies. Details of the theoretical calculations are described elsewhere [27–30]. The present experimental data is also compared with the DDCS data previously measured for the same beam energy [8].

However, detailed analysis and comparisons with the newly developed extensions of CDW-EIS model were absent there. Therefore, the present study of the DDCS as a function of the electron energy and scattering angle, the SDCS as well as the TCS is important for the validity check of the theoretical predictions for such multi-electron systems like $CH_4$ in collisions with such low-energy projectile beams.

## 2. Experimental Technique

Details of the experimental techniques are described elsewhere [3,31]. In brief, an ECRIA (electron cyclotron resonance-based ion accelerator) was used to obtain the proton beams. Initially, hydrogen plasma was created in the plasma chamber, with the help of 14.5 GHz microwaves. The plasma is contained in the plasma chamber with the help of axial and radial magnetic fields generated by the permanent magnets. An extraction voltage of 30 keV is applied after the plasma chamber to extract the proton ions produced in the plasma chamber.

The ECR platform was raised to 220 kV to obtain a 250 keV $H^+$ ion beam, and the ions are further accelerated by the accelerating columns followed by an electrostatic quadrupole triplet lens to focus the ion beam with X–Y deflectors to control the beam direction. A switching magnet directs the beam to the desired beam line. Another quadrupole triplet lens is used after the switching magnet for further focusing of the beam in the scattering chamber. Two sets of four jaw slits kept apart are used to cut the beam size and make an approximately parallel beam.

The well-collimated beam is passed through a two mm aperture with a length of 30 mm, and finally ions are made to go through a 16 mm long collimator of 4 mm diameter, which opens up at the entrance of the scattering chamber with a diameter of 6 mm to control the well-collimated beam without any further scattering. Low-energy electrons can be deflected even by a small magnetic field. To reduce the earth's magnetic field, we use $\mu$ metal shielding inside the chamber to increase the collection efficiency of the low-energy electrons. The projectiles interact with methane gas in a flooded chamber at a static pressure of 0.08 mT.

The ejected electrons are energy analyzed using a hemispherical energy analyzer and are detected by a channel electron multiplier (CEM). A pre-acceleration voltage of 6 V is applied to the entrance and exit slits of the analyzer to reduce the loss of low-energy electrons. The dependence of the yield of electrons on the pre-acceleration voltage was studied, and we found that, at and above 6 V, the cross section is practically saturated. The front of the CEM was biased to +100 V in order to detect the lower energy electrons with higher efficiency. The efficiency of the detector was found to be 85% (over the energy range studied) after putting the voltage in the front of the CEM.

The data were collected for the energy range of 1–360 eV and at scattering angles between 20° and 160°. The signal from CEM ($\sim$100 mV) is amplified by a 'ORTEC 474' timing filter amplifier (TFA) for the amplification of the signal to a value of $\sim$1 V. A constant fraction discriminator (CFD) was used to reduce the noise from the main signal by setting a threshold. At the last stage, a level translator switches the negative CFD signal to the positive one, and finally the data is stored in a computer with the help of the 'Labview' software through the National Instruments (NI) modules. The total error in the data is $\sim$17% considering the statistical fluctuations, errors in pressure variation, solid angle path length, detector efficiency and energy resolution of the analyzer.

## 3. Results and Discussions

### 3.1. Energy and Angular Distributions

In Figure 1a–f, we display the DDCS for six different angles: 20°, 30°, 60°, 90°, 120° and 160°, respectively. The present as well as existing data sets are shown in Tables 1–4. The measured data is compared with the different CDW-EIS approaches, such as CNDO, MO ($d = 0.7$) and MO ($d = 1.0$) models. The CNDO model considers the molecular orbital as a sum of the constituent atomic orbitals. In the MO approach, single center character of the molecule was considered using two different inter-nuclear distance scaling parameters $d = 0.7$ and 1.0. More detailed descriptions of the models are given elsewhere (see Section 2 in [27]).

The large cross section at a low-energy region of the spectra arises from the soft collision process where the projectile ion interacts with the electrons at a large impact parameter. The two-center effect and the binary-encounter process contribute to the intermediate

energy region. The carbon KLL Auger peak is observed at 240 eV for backward angles. The DDCS data shows that, below 10 eV, the CDW-EIS(MO) results with a scaling factor of $d = 0.7$ as well as the CNDO model better match the measured data. The MO model with $d = 1$ underestimates the data at low-energy regions for all angles; however, above 10 eV, it matches with the MO model with $d = 0.7$ and works better than the CNDO calculations for the total energy range.

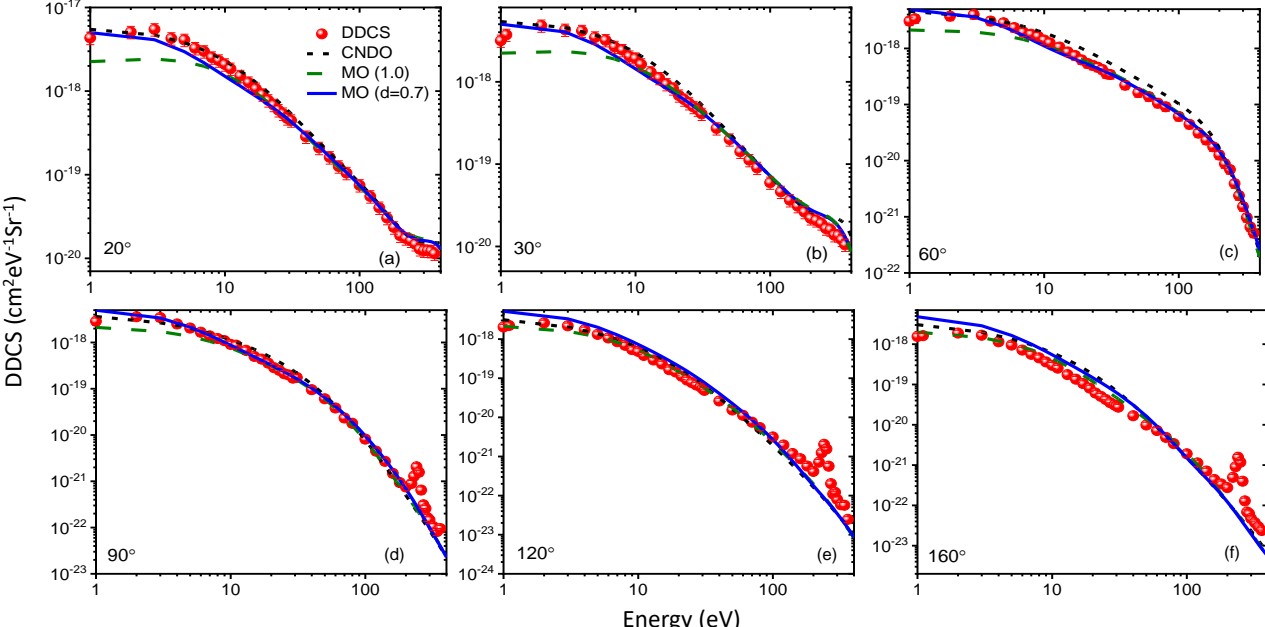

**Figure 1.** Energy distributions of DDCS (red circles) along with different theoretical calculations for several scattering angles. The solid, dashed and dotted lines are CDW−EIS (MO) ($d = 1$), CDW−EIS (MO) ($d = 0.7$) and CDW−EIS (CNDO), respectively.

The angular distributions of the DDCS (Figure 2) are represented for different electron energies. At lower energy, the distribution was found to be almost isotropic. However, with an increase in energy, the forward–backward asymmetry increases. This asymmetry arises when the velocity of the ejecting electron increases such that the difference with the velocity of the projectile decreases, and the cross section for forward angles begins to increase with respect to the cross section at backward angles.

Therefore, the angular distribution of electron DDCS falls sharply with the angle. The forward–backward difference is large compared to MeV energy collisions. For the lower velocity of the projectile, the interaction time with the target molecule is longer. Due to this reason, the projectile ions can drag the secondary scattered electrons towards the forward direction along with the projectile itself. Due to this post collisional effect, the forward–backward asymmetry is larger in the case of a low-velocity projectile than it is for a projectile with higher velocity.

The results agree with the previously published data [8]. However, a closer look reveals differences in the forward angles. The CDW-EIS (MO) model ($d = 1$) works better in most of the cases. The CDW-EIS (CNDO) differs most from the data at low energies. A hump-like structure, mostly due to binary collision, is seen for high electron energies (>50 eV), as shown in Figure 2d,e at around 60°, which was also predicted by the calculations.

**Table 1.** DDCS data in collisions with 250 keV $H^+$ in units of $10^{-18}$ cm$^2$eV$^{-1}$sr$^{-1}$ (uncertainty $\sim$ 17%).

| Energy (eV) | 20° | 30° | 45° | 60° | 75° | 90° | 105° | 120° | 135° | 150° | 160° |
|---|---|---|---|---|---|---|---|---|---|---|---|
| 1 | 4.33 | 3.16 | 3.19 | 3.07 | 3.04 | 2.88 | 2.26 | 2.03 | 1.92 | 1.76 | 1.55 |
| 3 | 5.47 | 4.29 | 3.82 | 4.01 | 3.71 | 3.46 | 2.53 | 2.19 | 2.02 | 1.73 | 1.67 |
| 7 | 2.98 | 2.66 | 2.28 | 1.95 | 1.78 | 1.39 | 1.07 | 0.859 | 0.684 | 0.601 | 0.559 |
| 11 | 1.85 | 1.63 | 1.45 | 1.26 | 1.07 | 0.876 | 0.573 | 0.382 | 0.330 | 0.272 | 0.254 |
| 15 | 1.26 | 1.11 | 0.979 | 0.863 | 0.729 | 0.499 | 0.309 | 0.232 | 0.148 | 0.135 | 0.136 |
| 21 | 0.775 | 0.682 | 0.645 | 0.539 | 0.432 | 0.286 | 0.162 | 0.113 | 0.073 | 0.069 | 0.061 |
| 31 | 0.445 | 0.408 | 0.366 | 0.341 | 0.241 | 0.176 | 0.059 | 0.049 | 0.036 | 0.031 | 0.027 |
| 40 | 0.282 | 0.272 | 0.238 | 0.221 | 0.166 | 0.096 | 0.036 | 0.026 | 0.019 | 0.018 | 0.017 |
| 50 | 0.209 | 0.199 | 0.188 | 0.163 | 0.114 | 0.061 | 0.021 | 0.015 | 0.011 | 0.011 | $9.86 \times 10^{-3}$ |
| 60 | 0.161 | 0.141 | 0.142 | 0.138 | 0.087 | 0.038 | 0.013 | 0.011 | $7.62 \times 10^{-3}$ | $7.48 \times 10^{-3}$ | $7.20 \times 10^{-3}$ |
| 70 | 0.125 | 0.112 | 0.114 | 0.106 | 0.065 | 0.023 | $9.15 \times 10^{-3}$ | $7.52 \times 10^{-3}$ | $5.53 \times 10^{-3}$ | $5.76 \times 10^{-3}$ | $4.87 \times 10^{-3}$ |
| 80 | 0.105 | 0.090 | 0.109 | 0.091 | 0.049 | 0.018 | $6.68 \times 10^{-3}$ | $5.47 \times 10^{-3}$ | $3.85 \times 10^{-3}$ | $3.76 \times 10^{-3}$ | $3.40 \times 10^{-3}$ |
| 100 | 0.075 | 0.059 | 0.073 | 0.061 | 0.020 | $8.16 \times 10^{-3}$ | $3.79 \times 10^{-3}$ | $3.15 \times 10^{-3}$ | $2.06 \times 10^{-3}$ | $2.15 \times 10^{-3}$ | $1.91 \times 10^{-3}$ |
| 120 | 0.055 | 0.046 | 0.059 | 0.044 | 0.011 | $4.48 \times 10^{-3}$ | $2.07 \times 10^{-3}$ | $1.96 \times 10^{-3}$ | $1.41 \times 10^{-3}$ | $1.21 \times 10^{-3}$ | $1.13 \times 10^{-3}$ |
| 140 | 0.040 | 0.036 | 0.051 | 0.031 | $6.13 \times 10^{-3}$ | $2.70 \times 10^{-3}$ | $1.25 \times 10^{-3}$ | $1.19 \times 10^{-3}$ | $8.24 \times 10^{-4}$ | $8.11 \times 10^{-4}$ | $7.11 \times 10^{-4}$ |
| 160 | 0.030 | 0.031 | 0.039 | 0.023 | $3.31 \times 10^{-3}$ | $1.48 \times 10^{-3}$ | $9.01 \times 10^{-4}$ | $8.76 \times 10^{-4}$ | $4.81 \times 10^{-4}$ | $5.26 \times 10^{-4}$ | $4.39 \times 10^{-4}$ |
| 180 | 0.023 | 0.026 | 0.032 | 0.018 | $1.19 \times 10^{-3}$ | $9.38 \times 10^{-4}$ | $6.48 \times 10^{-4}$ | $5.68 \times 10^{-4}$ | $3.66 \times 10^{-4}$ | $4.08 \times 10^{-4}$ | $3.40 \times 10^{-4}$ |
| 200 | 0.019 | 0.021 | 0.028 | 0.013 | $1.51 \times 10^{-3}$ | $7.53 \times 10^{-4}$ | $4.91 \times 10^{-4}$ | $4.14 \times 10^{-4}$ | $3.33 \times 10^{-4}$ | $3.27 \times 10^{-4}$ | $2.76 \times 10^{-4}$ |

**Table 2.** DDCS data taken from the experiment of Lynch et al. [8] (scanned from Figure 6) in units of $10^{-18}$ cm$^2$eV$^{-1}$sr$^{-1}$.

| Energy (eV) | 15° | 20° | 30° | 40° | 50° | 60° | 70° | 80° | 90° | 110° | 125° |
|---|---|---|---|---|---|---|---|---|---|---|---|
| 10 | 2.38 | 1.97 | 1.50 | 1.31 | 1.11 | 1.03 | 0.966 | 0.873 | 0.662 | 0.540 | 0.440 |
| 20 | 0.873 | 0.934 | 0.844 | 0.662 | 0.662 | 0.578 | 0.522 | 0.440 | 0.324 | 0.175 | 0.116 |
| 50 | 0.309 | 0.298 | 0.277 | 0.277 | 0.240 | 0.223 | 0.192 | 0.115 | 0.069 | 0.024 | 0.019 |
| 100 | 0.138 | 0.103 | 0.092 | 0.103 | 0.103 | 0.086 | 0.051 | 0.021 | $7.65 \times 10^{-3}$ | $3.54 \times 10^{-3}$ | $3.06 \times 10^{-3}$ |
| 200 | 0.028 | 0.030 | 0.033 | 0.039 | 0.029 | 0.014 | $3.67 \times 10^{-3}$ | $1.31 \times 10^{-3}$ | $7.07 \times 10^{-4}$ | $4.71 \times 10^{-4}$ | $4.07 \times 10^{-4}$ |

**Table 3.** Energy and angular distributions of SDCS in units of $10^{-18}$ cm$^2$eV$^{-1}$ and $10^{-18}$ cm$^2$sr$^{-1}$, respectively (uncertainty $\sim$17%).

| Energy (eV) | SDCS ($d\sigma/dE$) | Angle (Degree) | SDCS ($d\sigma/d\Omega$) |
|---|---|---|---|
| 1 | 33.04 | 20 | 71.22 |
| 3 | 39.35 | 30 | 64.38 |
| 7 | 18.47 | 45 | 59.05 |
| 11 | 10.93 | 60 | 51.06 |
| 15 | 6.97 | 75 | 40.27 |
| 21 | 4.16 | 90 | 30.64 |
| 31 | 2.35 | 105 | 20.46 |
| 40 | 1.50 | 120 | 16.73 |
| 50 | 1.07 | 135 | 13.88 |
| 60 | 0.806 | 150 | 11.99 |
| 70 | 0.610 | 160 | 11.38 |
| 80 | 0.512 | | |
| 100 | 0.315 | | |
| 120 | 0.228 | | |
| 140 | 0.171 | | |
| 160 | 0.129 | | |
| 180 | 0.101 | | |
| 200 | 0.082 | | |

**Table 4.** Measured total cross section (TCS) of methane along with the values obtained from theoretical predictions in $10^{-16}$ cm$^2$ (uncertainty $\sim$17%). (* Interpolated value from [5]).

| Projectile | TCS$_{Exp}$ | TCS$_{Lynch}$ | TCS$_{Rudd}$ | TCS$_{CNDO}$ | TCS$_{MO(d=1)}$ | TCS$_{MO(d=0.7)}$ |
|---|---|---|---|---|---|---|
| 250 keV $H^+$ | 4.27 | 3.10 | 4.92 * | 5.81 | 3.92 | 5.37 |

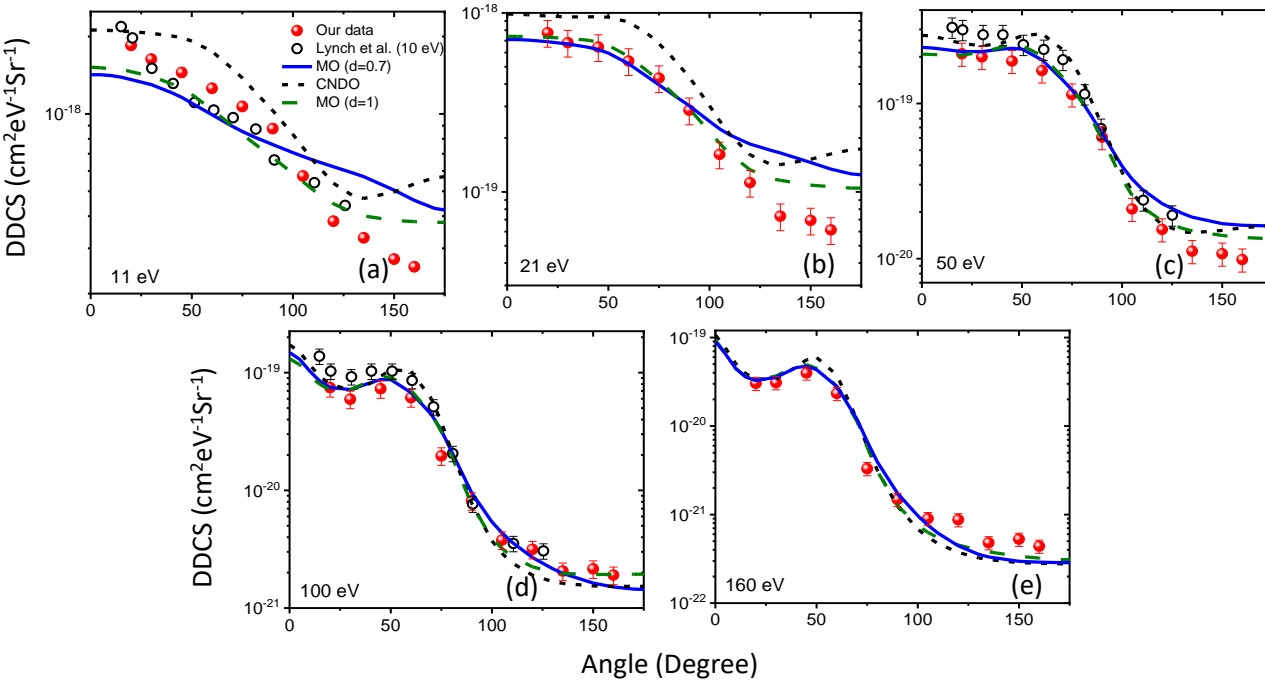

**Figure 2.** Angular distributions of the present DDCS (red circles) along with the models and the data available in the literature (black open circles) [8]. The solid, dashed and dotted lines are CDW−EIS (MO) ($d = 1$), CDW−EIS (MO) ($d = 0.7$) and CDW−EIS (CNDO), respectively.

### 3.2. Single Differential Cross Section (SDCS) and Total Cross Section (TCS)

The energy (E) distribution of the SDCS ($d\sigma/dE$) shown in Figure 3a was obtained by integrating the DDCS over the emission angles. Through integrating the DDCS over the total energy region, the angular distribution of SDCS ($d\sigma/d\Omega$) was derived and is shown in Figure 3b. Among all three calculations, the MO ($d = 0.7$) model matches better with the experimental results at the forward angles. The CNDO model overestimates the data, and MO ($d = 1$) diverges from the measured data at lower energy below 10 eV although it matches better with the measured value for backward angles. For the angular distribution of SDCS, a peak-like structure is predicted in the three model calculations around 60°, which is not visible in the experimental data, although such a structure was visible in the DDCS angular distributions.

The total ionization cross section was derived by integrating the DDCS over the energy range of 1–360 eV and the emission angles over 0–180°. The present data is compared with the existing data sets [5,8]. We used the TCS from Lynch et al. [8]. The present results (TCS = 427 Mb) were found to be closer to the CDW-EIS (Mo, $d = 1$) prediction, i.e., 392 Mb, whereas, those from Lynch et al., i.e., 310 Mb [8], showed large deviations from the theoretical models.

However, the linearly interpolated value of TCS obtained from the measurements of Rudd et al. [5] showed deviations from the present data and from Ref. [8] and, thus, deviates the most from the value calculated by the MO ($d = 1.0$) model. The present measurement is relatively closer to the CDW-EIS (Mo, $d = 1$) model. The CDW-EIS(MO, $d = 0.7$) also works well but the TCS calculated with the CNDO model overestimates the data most. We believe that the presently measured data are an improvement over those published earlier [5,8]

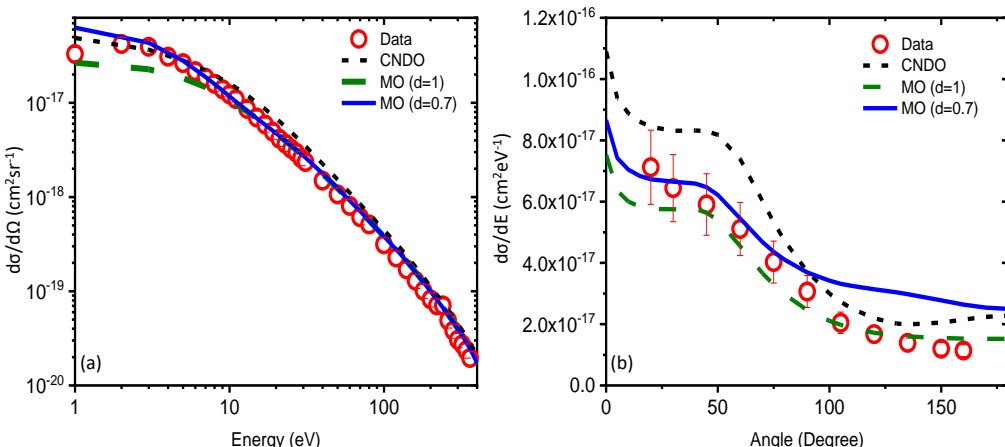

**Figure 3.** a and b represent the energy and angular distribution of SDCS, respectively, along with the theoretical models.

### 4. Conclusions

We have made detailed measurements of the electron DDCS for 250 keV protons colliding with the $CH_4$ molecule. The energy distributions as well as the angular distributions of the low-energy electrons were measured. The angular distributions were also compared with the existing data [8] in a few cases. The carbon KLL Auger electron peak was observed at 240 eV for the backward angles. The present data have better agreement with the different CDW-EIS model calculations, which were developed using a MO description of the target molecule.

The SDCS and TCS were deduced from the DDCS data and compared with the theoretical models. We found that the CDW-EIS (MO) model, with scaling parameter $d = 1.0$, was in better agreement with the DDCS, SDCS and TCS when compared to the CDW-EIS (CNDO) approach. We also found that the present TCS data provided better matching with the above model (MO with $d = 1.0$) compared to both the earlier measurements. The present study helps to validate the CDW-EIS model in dealing with a many-electron system, such as methane, at relatively low-energy (keV) collisions.

**Author Contributions:** L.C.T. conceived the idea of the experiment. L.C.T. and D.C. executed the experiment, analysis and interpretation. L.G. performed the theoretical calculations. D.C. and L.C.T. wrote the paper in consultation with all authors. All authors have read and agreed to the published version of the manuscript.

**Funding:** The support of the Department of Atomic Energy, Government of India, under project No. 12P-R&D-TFR-5.02-0300 is acknowledged. L.G. acknowledges the support from the Hungarian Scientific Research Fund (Grant No. K 128621).

**Institutional Review Board Statement:** Not applicable.

**Informed Consent Statement:** Not applicable.

**Data Availability Statement:** Not applicable.

**Acknowledgments:** We would like to acknowledge Nilesh Mhatre, W. A. Fernandes, Devendra Pathare, Thulasi Ram and S. N.Manjarekar for their technical support during the experiment.

**Conflicts of Interest:** The authors declare no conflict of interest.

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
