# Peer review of "Electron Emission Cross Section from Methane under 250 keV Proton Impact"

_atoms, doi:10.3390/atoms11030049_

Round 1
Reviewer 1 Report
In this manuscript, the authors show the DDCS of emission electron in the collision of 250 keV proton with CH4 both in experimental and theoretical. The result shows a mild oscillation which due to the Cohen-Famo type interference as they claimed. The measurement is reliable and the data is high quality. I think the manuscript could published in Atoms. To improve the manuscript, I have one concern that it will be easy to understand the electron emission pattern for reader if authors show the Schematic of the experiment arrange. I think the description about the experiment is too simple.
Author Response
Thank you for your valuable comments. Please find the replies attached below.

Reviewer 2 Report
The manuscript presents double and single differential cross sections and total cross sections of electrons emitted from CH4 molecule in collisions with 250 keV protons. The experimental data are compared to improved CDW-EIS model calculations. Scaling factors are added to the calculations to better adjust to the experimental data. No physical conclusions are derived from the comparison between the exp. data and model calculations. The discussion is mainly quantitative. From the exp. data, Young-type interference structures are considered, but need to be better evaluated. or presented. Some conclusions are not properly presented. (ex. contributions of the pairs).
Suggestions are included in text. Some sentences need to be rewritten.

Author Response
Thank you for your valuable suggestions. Please find the attached file with our reply.

Round 2
Reviewer 2 Report
Dear Editor
The authors have adequately addressed all my comments and concerns. They have answered the technical questions. The manuscript has been significantly improved after revising.
I recommend the manuscript for publication in the present form.
best regards
Author Response
Thank you for thoroughly reviewing the paper. We have replied to all the points mentioned by the referee. According to the referee the paper should be publishable in the last submitted form. But the word count was less to make it an article. Therefore, we have made some changes to the paper to make it suitable for a communication. Those are marked in bold and mentioned in the letter to the editor.